# Reconstructive Surgery versus Primary Closure following Vulvar Cancer Excision: A Wide Single-Center Experience

**DOI:** 10.3390/cancers14071695

**Published:** 2022-03-26

**Authors:** Mustafa Zelal Muallem, Jalid Sehouli, Andrea Miranda, Helmut Plett, Ahmad Sayasneh, Yasser Diab, Jumana Muallem, Imad Hatoum

**Affiliations:** 1Department of Gynecology with Center for Oncological Surgery, Charité Universitätsmedizin Berlin, Corporate Member of Freie Universität Berlin, Humboldt-Universität zu Berlin, and Berlin Institute of Health, Virchow Campus Clinic, Charité Medical University, 13353 Berlin, Germany; jalid.sehouli@charite.de (J.S.); andrea.miranda@charite.de (A.M.); pletth@googlemail.com (H.P.); jumana.muallem@charite.de (J.M.); imadgyn@yahoo.com (I.H.); 2Department of Gynecological Oncology, Surgical Oncology Directorate, Guy’s and St Thomas’ NHS Foundation Trust, Faculty of Life Sciences and Medicine, School of Life Course Sciences, King’s College London, Westminster Bridge Road, London SE1 7EH, UK; ahmad.sayasneh@gstt.nhs.uk; 3Department of Gynecology, Portland Hospital, Portland, VIC 3305, Australia; ydiab.pdh@swarh.vic.gov.au

**Keywords:** vulvar cancer, reconstructive surgery, wound healing, primary closure, vulvectomy, flap

## Abstract

**Simple Summary:**

When it comes to advanced vulval cancer management, there is a critical quandary to consider. This is owing to the severe negative impact of demolitive surgery on women who are afflicted by both functional and psychological consequences of the procedure. Primary closure of vulvar and/or perineal defects can be accomplished without difficulty in many situations, but this is accompanied by tension of the skin closure and distortion of the anatomy. In these circumstances, reconstructive surgery will be required to restore the anatomical and functional characteristics of the vulva. In this paper, we share our substantial expertise of primary closure versus reconstruction after demolitive surgery of advanced vulvar cancer, and we discuss our findings in light of the literature.

**Abstract:**

(1) Background: plastic reconstruction in vulvar surgery can lead to a better treatment outcome than primary closure. This study aims to compare the preoperative parameters (co-morbidities and tumor size) and postoperative results (tumor free margins and wound healing) between the primary closure and reconstructive surgery after vulvar cancer surgery; (2) Methods: this is a retrospective analysis of prospectively collected data from 2009 to 2021 at a tertiary cancer institution; (3) Results: 177 patients were included in the final analysis (51 patients had primary closure PC and 126 had reconstructive surgery RS). About half (49%) of the PC patients had no co-morbidities (*p* = 0.043). The RS group had a 45 mm median maximal tumor diameter compared to the PC group’s 23 mm (*p* = 0.013). More than 90% of RS and 80% of PC had tumor-free margins (*p* = 0.1). Both groups had anterior vulvar excision as the most common surgery (52.4% RS vs. 23.5% PC; *p* = 0.001). Both groups had identical rates of wound healing disorders. In a median follow-up of 39 months; recurrent disease was found in 23.5% of PC vs. 10.3% in RS (*p* = 0.012). In terms of overall survival there was no significant difference between the both groups; (4) Conclusions: reconstructive vulvar surgery enables enhanced complete resection rates of larger vulvar tumors with better anatomical restoration and a comparable wound recovery in comparison to primary closure. This results in a lower recurrence rate despite the increased tumor volume.

## 1. Introduction

According to Globocan, approximately 42,240 women worldwide will be diagnosed with vulvar cancer in 2020. The risk of this malignancy is higher in Europe and North America [1]. The primary treatment for vulvar cancer is surgery (except for patients who cannot undergo surgery because of their medical background or if they have extensive or metastatic disease). Surgery has become increasingly tailored for small tumors (sentinel lymph node staging and smaller resected free margins) [2,3]. However, in cases with advanced or multifocal tumors, more radical surgery is often required, and in cases of extensive malignancies, a total radical vulvectomy or an exenteration may become the treatment of choice.

Even if primary wound edge adaptation is usually feasible, the effects of bringing the wound edges together are neither visually nor functionally satisfying. Furthermore, creating skin pockets makes good aftercare more difficult in these instances. Plastic vulva restoration is essential, especially in cases of total vulvectomy, to restore the woman’s sense of physical wholeness. Reconstructive surgery can preserve the patient’s self-esteem and sense of femininity when conducted to restore both the vulvar architecture and function [4,5,6,7].

Following extensive excision of the vulvar lesion and surrounding skin, the critical steps to achieve a good reconstruction are the skin closure with high-quality tissues and preservation of the vaginal and urethral introitus with no shrinkage or displacement from their central position. If necessary, the anovaginal partition can be restored. An exenteration or abdominoperineal excision might weaken pelvic support, necessitating the filling-in of a variable amount of dead space. Secondary objectives will include sensation and sexual function restoration, satisfying external refurbishment, and avoiding the flap donor site morbidity [8].

This study aims to analyze the preoperative parameters for vulvar excisional surgery including co-morbidities and tumor size, and postoperative results including tumor free margins and wound healing. It also aims to compare these parameters, the surgical results and the oncological outcomes between primary closure and reconstructive surgery following vulvar cancer excision.

## 2. Materials and Methods

Our department of Gynecology with center for Oncological Surgery, Charité Medical University of Berlin conducted this retrospective analysis of prospective collected data from primary vulvar cancer patients who underwent surgery between January 2009 and December 2021. Only participants with recurring vulvar cancer and those with tumors that were less than 5 mm horizontally in diameter were excluded from the study. The Charité Medical University’s institutional review board gave its approval to this research (EA4/115/15). Before any clinical data were collected, written informed consent was obtained from each patient. The characteristics of the patient and operation were analyzed including the patient’s age at surgery, co-morbidities (as determined by Charlson’s co-morbidity index [9,10] and the American Society of Anesthesiologists (ASA) physical status scale [11]), histological type of vulvar tumor, grading, tumor stage as determined by the 2009 FIGO classification [12], involvement of inguinal lymph nodes, tumor size, anatomical localization, type of demolitive and reconstructive surgery, and the postoperative complications. Tumor-free margins were defined as pathological margins of at least 3 mm clear of cancer. The first author performed the majority of the reconstructive procedures utilizing one of the following flap options: fasciocutaneous V-Y flap, anterior pedicle labial flap, posterior pedicle labial flap, limberg flap, or myocutaneous flaps. In two individuals, the pelvic floor was reconstructed following a posterior exenteration by employing the corpus uteri as a muscle flap (Figure 1) [13].

Throughout the past four years, intraoperative indo-cyanine green angiography (SPY-Portable Handheld Imager, SPY-PHI, Stryker, Kalamazoo, MI, USA) was utilized to visualize blood flow in arteries and concomitant tissue perfusion (2018–2021). (Figure 2 depicts a fluorescence imaging system perfusion assessment intraoperative after indocyanine green injection and the necrotic flap one week following surgery.)

The Charité Medical University Berlin conducted the statistical analysis. IBM SPSS Statistics 21.0 was used to run all analyses (SPSS, Chicago, IL, USA). Descriptive statistics were used to analyze the data. Categorical data were de-scribed using frequency counts and percentages, whereas continuous variables were summarized using the median and range. A *p* value of 0.05 was used to determine statistical significance, and two-sided tests were used. Overall survival was estimated from the day of primary surgery to the last day of follow-up or death from any cause (event) (censored). Progression-free survival was assessed from the day of main surgery until the cancer recurrence or death from any cause. For progression-free and overall survival, Kaplan–Meier curves were created for both groups (PC vs. RS).

## 3. Results

This study comprised 177 patients with vulvar cancer who had primary surgery and were included in the final analysis. 51 patients underwent primary closure (PC) of the wound edges, whereas 126 patients underwent reconstructive surgery (RS) with one or more flap types. The average age of patients at the time of their initial diagnosis was 70 years (range: 28–91). A Charlson’s co-morbidity score of zero indicated that two-thirds (66.7 percent) of patients in the RS group did not have any co-morbidities, but fewer than half (49 percent) of patients in the PC group did not have any co-morbidities (*p* = 0.043). More than twice as many patients in the PC group had a high Charlson’s co-morbidity score (>3) as those in the RS group. However, the similar tendency was observed in the ASA-physical status scale study, albeit the difference was not statistically significant (*p* = 0.582). When reconstructive surgery was included, the operation time increased by an average of one hour to 134 (65–335) minutes in the RS group, as opposed to 67 (20–280) minutes in the PC group (*p* = 0.001). Histopathology investigation revealed squamous cell carcinoma in 158 (89.3%) cases, and moderately differentiated cancer in more than two-thirds of the cases. The median maximum diameter of the resected tumor was 45 mm (20–127) in the RS group compared to 23 mm (8–89) in the PC group (*p* = 0.013). 

The demographic and pathology characteristics are summarized in Table 1.

The most common type of excisional surgery in both study groups was the anterior vulvar excision, which was performed in 52.4% in the RS group vs. 23.5% in the PC group (*p* = 0.001), followed by the radical vulvectomy in 19% in the RS group vs. 15.7% in the PC group (*p* = 0.584). Other procedures included posterior vulvar excision (12.4%), left hemi-vulvectomy (8.5%), right hemi-vulvectomy (6.7%), wide local excision (7.9%) which was only indicated in the PC group, and Exenteration (2.3%) which was only indicated in the RS group. In the entire group, clitoris resection was required in 65% of patients, although it was much more common in the RS group with 93 patients (73.8%) compared to 22 patients (43.1%) in the PC group (*p* < 0.000). The urethra was resected partially or completely as part of the demolitive operation in 32 patients (25.4%) in the RS group and just 5 patients (9.8%) in the PC group (*p* = 0.035). In three cases, a total urethral resection was performed and a permanent supra-pubic catheter was required in two of them. In the third case, a complete bladder resection with an ileum-conduit was the treatment of choice. Only in the RS group, partial or total anus resection was indicated in 14 cases (11.1%). Creating an anus praeter (sigmoid stoma) was required in three cases of the total anus resection (posterior exenteration).

Sentinel lymph node staging was used in 56.5% of cases. The inguinal lymph node dissection was conducted in 41.2% of patients. Seventy-one percent of lymph nodes in the RS group and 74.6% of lymph nodes in the PC group (*p* = 0.255) were not affected.

When selecting how to proceed with reconstructive surgery, the location, shape, and volume of the defect were all taken into account. More than half (54 percent) of reconstructive surgeries were performed using fasciocutaneous V-Y flaps, while for the remaining (11%) we used anteroposterior or posteroposterior pedicle labial or a Limberg flap in one or both locations. Myocutaneous flaps were used in a total of 20 patients (15.9%), while the corpus uteri flap was used in two patients (1.6%). In 22 patients (17.5%) a flap combination was deemed necessary. More than 90% in the RS group and more than 80% of patients in the PC group had tumor-free margins (114 patients and 41 patients, respectively). R1-status patients had to undergo further surgery in order to reach R0-status in most situations. The wound healing disorder (Grad III according to Calvin-Dindo classification [14]) was registered in 16 patients (12.7%) of RS group vs. seven patients of PC group (13.7%), (*p* = 1). We reported a case of a total flap failure in the RS group (Figure 2). All remaining cases of wound healing disorder in the recipient site were wound breakdown. In these cases, a second reconstructive surgery was requited. In 32 of the cases, additional treatment (radio- or radiochemotherapy) was required. 

The demolitive and reconstructive surgery characteristics are summarized in Table 2.

In a median follow-up of 39 months, recurrent disease was found in 12 PC group patients (23.5%) and 13 RS group patients (10.3%) (*p* = 0.012). During the above-mentioned follow-up period, 11 patients (21.6%) from the PC group and 29 patients (23%) from the RS group died (*p* = 0.992). The Kaplan-Meier analyses of progression-free and overall survival are demonstrated in Figure 3.

## 4. Discussion

We have shown a slightly higher rate of complete resection with significant less need to revision surgery in the reconstructive surgery group compared to the primary closure (PC) patients, despite the fact that tumor volume, clitoris resection rate, and urethra and anus resection rates were much lower in the PC group. Both groups had the same rate of wound healing problem. When comparing the RS group to the PC group, the recurrence rate was considerably lower in the RS group. No significant differences in postsurgical complications or overall survival were found between the two groups. Other studies have found that plastic reconstruction after vulvectomy or extirpative surgery for vulvar cancer is related with a better aesthetic and functional outcome, as well as a decreased rate of wound healing issues.

Weikel et al. reported a greater primary healing rate, a shorter inpatient stay, and a better functional outcome following plastic reconstruction in a remarkably sizable trial comparing 103 plastic reconstructions to 110 primary wound closures [15]. Another study published in 2014 by Benedetti Panici et al. indicated that 29 reconstruction procedures (modified gluteal fold advancement V-Y flap) resulted in a shorter inpatient stay and a lower dehiscence rate (11% vs. 40%) when compared to 78 surgeries without plastic reconstruction. Clinical factors such as age, BMI, histological type, and FIGO staging did not differ statistically. Similarly, there was no statistically significant difference in terms of complications between the two groups. However, when only patients with tumors larger than 4 cm were considered (27 patients who received flap treatment vs. 30 patients who did not receive any flap treatment), complication rates were statistically reduced in those treated with reconstructive surgery [16].

According to the results of another retrospective study that compared 77 patients treated with direct closure with 72 patients treated with a reconstructive procedure, skin flap reconstruction decreases postoperative morbidity and provides better anatomical and functional results than direct closure of the perineal defect. Reconstructive operations reduced wound dehiscence to 26%, vaginal introitus stenosis to 2%, sexual dysfunction to 10%, and urine stream redirection to 1%, compared to 64%, 8%, 50%, and 5%, respectively, in the primary closure group according to the same study [17].

The main objective of the retrospective study conducted by Aviki et al. was to examine the margin status and prognostic factors for complications in patients who underwent vulvectomy for squamous cell carcinoma with and without plastic-assisted closure. The mean tumor volume was, comparable to our study, and considerably larger in the reconstruction group (3.73 cm versus 2.03 cm, *p* = 0.01). The study demonstrated that plastic aided closure, especially for tumors greater than 3 cm was substantially linked with acceptable margins and did not implicate problems [18].

Similar findings were achieved when Zahng et al. published a retrospective study of vulvar reconstruction using various types of flaps in 2015. In their conclusion, they stated that flap reconstruction is associated with a low rate of postoperative complications, lower discomfort, and an improved functional status [19].

An Italian group attempted to develop a comprehensive algorithmic strategy to vulvar cancer excision with surgical reconstruction, incorporating both standard and perforator flaps and considering anatomical subunits and defect form. They retrospectively analyzed 80 cases of vulvar cancer excisional surgery with repair conducted between June 2006 and January 2016, and a total of 101 flaps. They created an algorithm based on their experience to aid tselecting the flap for vulvoperineal reconstitution following oncologic excisional surgery for vulvar cancer [8]. Apart from the knowledge of the plastic surgery team, we believe that implementing such an algorithm for identifying the best plastic flap is critical and might be beneficial in many circumstances.

In this study, we have not discussed the vast options of using the perforator flaps for reconstructing the vulvoperineal defects, as these flaps did not exist in our study. The latter flaps, however, are well known to be very effective because of their low complication rate, minimal donor site morbidity, quick dissection, proximity of donor and recipient sites and the possibility to harvest large skin islands of variable thickness [20,21].

When addressing surgical treatment options following vulvar cancer excision, it is important to keep in mind the sexual and functional outcomes of reconstructive surgery in comparison to primary closure [22,23]. This component was not included in this paper, but it will be the focus of our near future research.

The strength of our study, that it includes, to the best of our knowledge, the largest number of patients underwent reconstructive surgery after f vulvar cancer excisin. The same team performed all surgeries, which minimize the interpersonal variation effect.

The main weakness in our study, however, is that it was conducted retrospectively at a single center. Although the literature search was successful in discovering relevant studies, it is difficult to make a conclusive statement or consensus due to the incomparability of the studies, the heterogeneous patient populations, and the varied study designs. Therefore, a prospective randomized trial may be needed to better elucidate the risks and benefits of vulvar reconstruction after cancer surgery.

## 5. Conclusions

Reconstructive procedures after primary excisional surgery for vulvar cancer may allow for the removal of larger vulvar tumorswith satisfactory anatomical restoration. The postoperative healing and complication rates are comparable to primary closure. The reconstructive surgery (RS) achieves a slightly higher rate of complete resection with significant less need to revision surgery than primary closure (PC). This results in considerably lower recurrence rate.

## Figures and Tables

**Figure 1 cancers-14-01695-f001:**
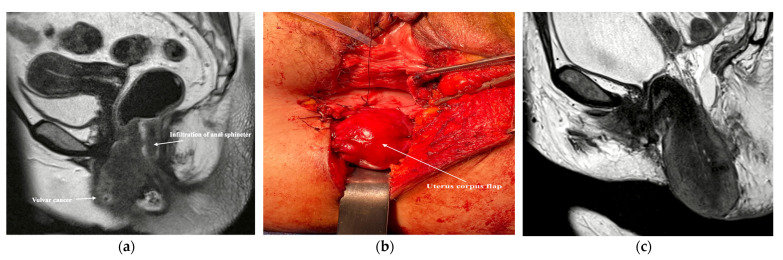
Pelvic reconstruction following a posterior exenteration with resection of Anus, posterior vaginal wall and vulva using the corpus uteri as a muscular flap: (**a**) the vulvar tumor with infiltration of anal sphincter; (**b**) intraoperative photo of adapting the corpus uteri flap to the defect edges after the demolitive surgery; (**c**) Magnetic resonance imaging of the corpus uteri flap two weeks after surgery.

**Figure 2 cancers-14-01695-f002:**
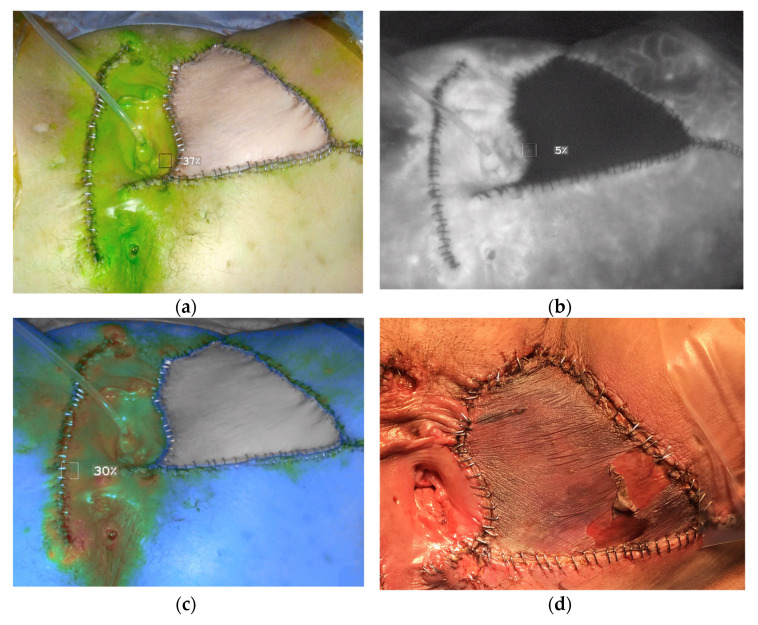
Perfusion assessment with fluorescence imaging system after injecting indocyanine green in (**a**) Overlay mode; (**b**) SPY fluorescence mode; (**c**) Color segmented fluorescence mode and (**d**) White light one week after surgery. This figure showed the very poor flap perfusion indicated with ICG-imaging study intraoperatively and the necrotic flap one week later.

**Figure 3 cancers-14-01695-f003:**
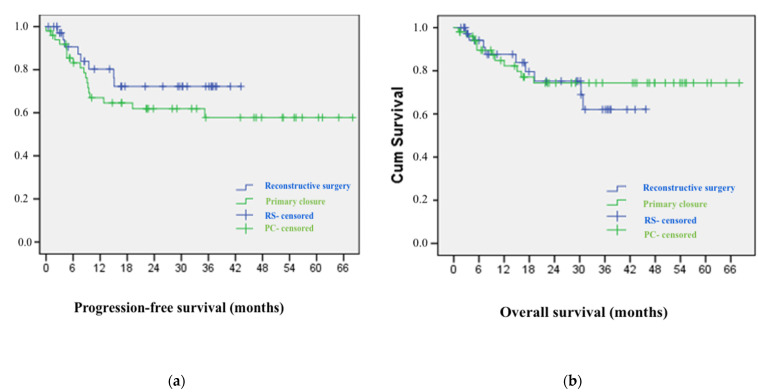
Kaplan-Meier curves of progression-free (**a**) and overall survival in both groups (**b**).

**Table 1 cancers-14-01695-t001:** Patient’s and tumor’s characteristics.

Characteristic		All Patients *n* = 177(%)	Reconstructive Surgery (RS) *n* = 126 (%)	Primary Closure (PC) *n* = 51 (%)	*p*-Value
Age at first diagnosis, median (range) years		70 (28–91)	71 (28–86)	67 (35–91)	0.604
Charlson’s comorbidity score	0	109 (61.6%)	84 (66.7%)	25 (49%)	0.043
	1–3	49 (27.7%)	32 (25.4%)	17 (33.3%)	
	4–6	17 (9.6%)	9 (7.1%)	8 (15.7%)	
	≥7	2 (1.1%)	1 (0.8%)	1 (2%)	
American Society of Anesthesiologists	ASA 1	19 (10.7%)	12 (9.5%)	7 (13.7%)	0.582
	ASA 2	111 (62.7%)	83 (65.9%)	28 (54.9%)	
	ASA 3	47 (26.6%)	31 (24.6%)	16 (31.4%)	
Histology	Keratinizing squamous	138 (78%)	97 (77%)	41 (80.4%)	0.768
	Non keratinizing squamous	20 (11.3%)	13 (10.3%)	7 (13.7%)	
	Basaloid	5 (2.8%)	4 (3.2%)	1 (2%)	
	Adenocarcinoma	14 (7.9%)	12 (9.5%)	2 (3.9%)	
Grading	G1	18 (10.2%)	10 (7.9%)	8 (15.7%)	
	G2	117 (66.1%)	81 (64.3%)	36 (70.6%)	
	G3	32 (18.1%)	26 (20.6%)	6 (11.8%)	
	unknown	10 (5.6%)	9 (7.1%)	1 (2%)	
Tumor volume, median (range) mm		31 (8–127)	45 (20–127)	23 (8–89)	0.013

**Table 2 cancers-14-01695-t002:** Demolitive and reconstructive surgery characteristics.

Characteristic		All Patients *n* = 177 (%)	Reconstructive Surgery (RS) *n* = 126 (%)	Primary Closure (PC) *n* = 51 (%)	*p*-Value
Duration of surgery, median (range) minutes		112 (20–335)	134 (65–335)	67 (20–280)	<0.001
Tumor free margins		155 (87.8%)	114 (90.4%)	41 (80.4%)	0.111
Wound healing disorders		23 (13%)	16 (12.7%)	7 (13.7%)	1.000
Type of demolitive surgery	Radical vulvectomy	32 (18.1%)	24 (19%)	8 (15.7%)	0.584
	Anterior vulvar resection	78 (44.1%)	66 (52.4%)	12 (23.5%)	0.001
	Posterior vulvar resection	22 (12.4%)	17 (13.5%)	5 (9.8%)	
	Hemivulvectomy left	15 (8.5%)	9 (7.1%)	6 (11.8%)	
	Hemivulvectomy right	12 (6.8%)	6 (4.8%)	6 (11.8%)	
	Wide excision	14 (7.9%)	0 (0%)	14 (27.5%)	
	Exentration	4 (2.3%)	4 (3.2%)	0 (0%)	
Clitoris resection		115 (65%)	93 (73.8%)	22 (43.1%)	0.000
Partial or total resection of urethra		37 (20.9%)	32 (25.4%)	5 (9.8%)	0.035
Partial or total resection of anus		14 (7.9%)	14 (11.1%)	0 (0%)	
Sentinel lymph node staging only		100 (56.5%)	74 (58.7%)	26 (50.1%)	
Inguinal lymph node dissection		73 (41.2%)	52 (41.3%)	21 (41.2%)	
Type of reconstructive surgery	Fasciocutaneous V-Y flap	-	54 (42.9%)	-	
	Anterior pedicle labial flap	-	8 (6.3%)	-	
	Posterior pedicle labial flap	-	11 (8.7%)	-	
	Limberg flap	-	9 (7.1%)	-	
	Myocutaneous flaps	-	20 (15.9%)	-	
	Flap combination	-	22 (17.5%)	-	
	Corpus uteri flap	-	2 (1.6%)	-	
FIGO	IA	21 (11.9%)	12 (9.5%)	9 (17.6%)	
	IB	79 (44.6%)	57 (45.2%)	22 (43.1%)	
	II	23 (13%)	18 (14.3%)	5 (9.8%)	
	IIIA	22 (12.4%)	16 (12.7%)	6 (11.8%)	
	IIIB	14 (7.9%)	12 (9.5%)	2 (3.9%)	
	IIIC	7 (4%)	4 (3.2%)	3 (5.9%)	
	IVA	9 (5.1%)	5 (4%)	4 (7.8%)	
	IVB	2 (1.1%)	2 (1.6%)	-	
Lymph node (LN) status	Not involved	127 (71.8%)	94 (74.6%)	33 (64.7%)	0.255
	1-2 LN with <5 mm	21 (11.9%)	13 (10.3%)	8 (15.7%)	
	>5 mm or >2 LNs or extra capsular infiltration	25 (14.1%)	19 (15.1%)	6 (11.8%)	
	No staging	4 (2.3%)	0 (0%)	4 (7.8%)	
Further therapy (Radio-/radiochemotherapy)		32 (18.1%)	23 (18.3%)	9 (17.6%)	

## Data Availability

The data presented in this study are available on request from the corresponding author. The data are not publicly available due to the privacy and ethical policy of our institution.

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
