# Peer review of "Reconstructive Surgery versus Primary Closure following Vulvar Cancer Excision: A Wide Single-Center Experience"

_cancers, 2022, doi:10.3390/cancers14071695_

Round 1

Reviewer 1 Report

This study does not seem to make an impact in which type of treatment is better unlike others in the field that have a clear stand although the sample size is not very different. The conclusion is weak. Having equal sample size may lead to better comparison and higher significant conclusions. 

Author Response

Thank you very much for reviewing our manuscript. To the best of our knowledge, this is the largest study about reconstructive surgery after primary excision of vulvar cancer, the study has even a long follow-up and provide survival data. This study clarified that reconstructive surgery (RS) patients had a slightly higher rate of complete resection with significant less need to revision surgery than primary closure (PC) patients, despite the fact that tumor volume, clitoris resection rate, and urethra and anus resection rates were much lower in the PC group. Both groups had the same rate of wound healing problem. When comparing the RS group to the PC group, the recurrence rate was considerably lower in the RS group. No significant differences in postsurgical complications or overall survival were found between the two groups. Other studies have found that plastic reconstruction after vulvectomy or extirpative surgery for vulvar cancer is related with a better aesthetic and functional outcome, as well as a decreased rate of wound healing issues.

This allowed us to conclude that: Reconstructive treatment after primary excisional surgery for vulval cancer can remove larger vulvar tumors with satisfactory anatomical restoration. The postoperative healing and complication rates are comparable to primary closure. The reconstructive surgery (RS) achieves a slightly higher rate of complete resection with significant less need to revision surgery than primary closure (PC). This results in considerably lower recurrence rate.

Reviewer 2 Report

The aim of this retrospective study is to compare the features and surgical results of primary closure versus reconstructive surgery after advanced vulvar cancer excision. They enrolled 177 patients with vulvar cancer at first diagnosis and with tumors more than 5 millimeters horizontally in diameter, underwent surgery between January 2009 and December 2021. 51 patients underwent primary closure (PC) of the wound edges and 126 patients underwent reconstruction surgery (RS) with one or more flap types. Flaps employed were fascio-cutaneous V-Y flap, anterior pedicle labial flap, posterior pedicle labial flap, limberg flap, or myo-cutaneous flaps. Results showed that more than 90% of RS and 80% of PC had tumor-free margins (p=0.1) with a recurrent disease found in 23.5% 36 of PC vs. 10.3% in RS (p=0.012). Both groups had identical rate of wound healing disorders.

The topic of this manuscript is interesting and timely.

In fact, in the Literature, several authors retrospectively showed their personal experience and results in flap reconstruction after excisional vulvar surgery, but there is no consensus. Common opinion is that flap reconstruction decreases postoperative morbidity and provides better anatomical and functional results than direct closure of the perineal defect.

Anyway, the manuscript needs major revisions before it can be considered for publication.

COMMENTS:

The language is poor throughout the text and the manuscript needs to be edited by a professional language editing service or by an English native speaker. Punctuation should be checked. Sentences such the following, make the manuscript difficult to read.

“after a considerable section of the lesion and neighboring skin has been removed, sskin closure with high-quality tissues and maintenance of vagina and urethral introitus without shrinking or displacement from their central position are the fundamental goals of reconstruction.”

In the Abstract the authors should provide more information:

  • In the background:
    • The aim of the study is not mentioned. Authors should rephrase abstract’s introduction.

  • In the Methods:
    • How were the patients enrolled? Did they adopt inclusion/exclusion criteria?
    • Which variables were included?
    • Which statistical tests were employed?
    • Some details about the surgical reconstructive technique, with particular reference to the flaps employed

  • In the Results
    • Some patients’ features such as BMI and Comorbidities
    • Preoperative radiotherapy and previous surgery

In the Main text:

In the “Introduction”:

  • Authors stated that “If necessary, the anovaginal partition can be restored.” I think the term “necessary” is incorrect, because the anovaginal partition is an important anatomical area which allow to separate the two orifice and it should be restore in any case whenever possible.
  • The purpose of the study should be better made explicit. Which kind of features and surgical results did authors refer to? Oncological outcome? Reconstructive outcome?

In the “Methods”

  • Post-operative complications recorded should be reported. Did they consider minor or major complications?
  • Which did variables determine the choice of reconstruction?
  • Authors stated that “In two individuals, the pelvic floor was reconstructed following a posterior exenteration by employing the corpus uteri as a muscle flap (Figure 1).” The employment of corpus uteri as a muscle flap is unclear in terms of safety, efficacy, and surgical technique. Can authors share more details citing the relevant references?
  • Did authors routinely use intraoperative indocyanine green angiography for each case of vulva reconstruction?
  • The flap showed in the figure n.2 seems to be completely necrotic according both to the fluorescence and clinical picture taken one week after surgery. This issue should be indicated in the caption
  • The authors should indicate postoperative management of these patient, if any.

In the “Results”

  • How many flaps were transplanted in the study? Which flaps were employed according to the type of excisional surgery (anterior vulvectomy, radical vulvectomy, posterior vulvectomy, hemi-vulvectomy, exenteration)?
  • Information about preoperative radiotherapy should be given, because this factor has a fundamental role in delayed wound healing and postoperative complications rate.
  • It is unclear the number of cases of total urethral resection. How did authors restore the case of partial urethral resection?
  • In which case did authors use myo-cutaneous flap? Which was the entity of the defect?
  • In figure 2, pictures revealed an intraoperative ischemic aspect of the flap using indocyanine green angiography. Why did authors decide to not harvest a second flap?
  • Authors should indicate how many patients with postoperative wound healing disorders needed a second-stage surgery.

In the Discussion:

  • Authors should indicate the strength and novelty of this retrospective study compared with others.
  • The importance of radiotherapy on complication rate should be better discussed, citing the relevant reference (PMID: 34830901)
  • The possibility of vulvoperineal reconstruction with perforator flaps should be mentioned and better discussed, citing the relevant references (PMID: 34866009; PMID: 27273808)

Author Response

Thank you very much for reviewing our manuscript and for this very good summary.

The topic of this manuscript is interesting and timely.

In fact, in the Literature, several authors retrospectively showed their personal experience and results in flap reconstruction after excisional vulvar surgery, but there is no consensus. Common opinion is that flap reconstruction decreases postoperative morbidity and provides better anatomical and functional results than direct closure of the perineal defect.

Anyway, the manuscript needs major revisions before it can be considered for publication.

The language is poor throughout the text and the manuscript needs to be edited by a professional language editing service or by an English native speaker. Punctuation should be checked. Sentences such the following, make the manuscript difficult to read.

“after a considerable section of the lesion and neighboring skin has been removed, sskin closure with high-quality tissues and maintenance of vagina and urethral introitus without shrinking or displacement from their central position are the fundamental goals of reconstruction.”

We reviewed the language by a professional language editing. This sentence is rewritten: ….

In the Abstract the authors should provide more information:

The abstract in Cancers has to be a single paragraph of about 200 words maximum, therefor we restricted die information given in abstract on the most important points.

In the background:

    • The aim of the study is not mentioned. Authors should rephrase abstract’s introduction.

We expended the background to include the aim of the study: plastic reconstruction in vulvar surgery can contribute to an improved treatment outcome compared to primary closure. This study aims to compare the preoperative parameters (comorbidities and tumor size) and postoperative results (tumor free margins and wound healing) between the primary closure and reconstructive surgery after vulvar cancer surgery.

  • In the Methods:
    • How were the patients enrolled? Did they adopt inclusion/exclusion criteria?
    • Which variables were included?
    • Which statistical tests were employed?
    • Some details about the surgical reconstructive technique, with particular reference to the flaps employed

Patients were enrolled in the department of Gynecology with center for Oncological Surgery, Charité Medical University of Berlin in the period between January 2009 and December 2021. In an all-commers design, consecutive patients with primary vulvar cancer, who primary operated, were included. Only participants with recurring vulvar cancer and those with tumors that were less than 5 millimeters horizontally in diameter were excluded from the study. The characteristics of the patient and operation were analyzed, including the patient's age at surgery, comorbidities (as determined by Charlson's comorbidity index and the American Society of Anesthesiologists (ASA) physical status scale), histological type of vulvar tumor, grading, tumor stage as determined by the 2009 FIGO classification, involvement of inguinal lymph nodes, tumor volume, anatomical localization, and type of demolitive and reconstructive surgery, and the postoperative complications. The first author performed the majority of the reconstructive procedures utilizing one of the following flap options: fasciocutaneous V-Y flap, anterior pedicle labial flap, posterior pedicle labial flap, limberg flap, or myocutaneous flaps. In two individuals, the pelvic floor was reconstructed following a posterior exenteration by employing the corpus uteri as a muscle flap. Descriptive statistics were used to analyze the data. Categorical data were de-scribed using frequency counts and percentages, whereas continuous variables were summarized using the median and range. P0.05 was used to determine statistical significance, and two-sided tests were used. Overall survival was estimated from the day of primary surgery to the last day of follow-up or death from any cause (event) (censored). From the day of main surgery until the cancer recurrence or death from any cause, progression-free survival was assessed. For progression-free and overall survival, Kaplan–Meier curves were created for both groups (PC vs. RS).

If we add only one sentence for every point mentioned above, the abstract will be very long and for sure not more accepted from the journal!

  • In the Results
    • Some patients’ features such as BMI and Comorbidities
    • Preoperative radiotherapy and previous surgery

A Charlson’s comorbidity score of zero indicated that two-thirds (66.7 percent) of patients in the RS group did not have any co-morbidities, but fewer than half (49 percent) of patients in the PC group did not have any co-morbidities (p=0.043). More than twice as many patients in the PC group had a high Charlson's comorbidity score (> 3) as those in the RS group. However, the similar tendency was observed in the ASA-physical status scale study, albeit the difference was not statistically significant (p=0.582).

All patients included in this study are primary operated patients. The primary radiotherapy followed with operation is not a standard in our hospital.

Again, adding all these information in abstract will result in a very long abstract, therefor we decided to add only the most important results.

In the Main text:

In the “Introduction”: 

  • Authors stated that “If necessary, the anovaginal partition can be restored.” I think the term “necessary” is incorrect, because the anovaginal partition is an important anatomical area which allow to separate the two orifice and it should be restore in any case whenever possible.

About 50% of our patients has no surgery in the posterior vulvar region (anovaginal partition), therefor we used if necessary. In these patients without surgery in anovaginal partition, there was no need for reconstruction the anovaginal partition.

  • The purpose of the study should be better made explicit. Which kind of features and surgical results did authors refer to? Oncological outcome? Reconstructive outcome?

We add this sentence to the end of introduction:
The purpose of this study is to analyze preoperative parameters, including comor-bidities and tumor size, and postoperative results, including tumor free margins and wound healing. And to compare these parameters, the surgical results and the oncological outcomes between primary closure and reconstructive surgery

In the “Methods”

  • Post-operative complications recorded should be reported. Did they consider minor or major complications?

The main postoperative complication after vulvar cancer surgery is the wound healing. We mentioned that the wound healing disorder (Grad III according to Calvin-Dindo classification [13]) was registered in 16 patients (12.7%) of RS group vs. 7 patients of PC group (13.7%), p=1.000. There was only one incidence of total flap loss in the RS group (figure 2 depicts the intraoperative assessment of flap perfusion with indocyanine green (a) and the post-operative ischemic flap(b)), and all other occurrences of wound healing disorder in the recipient site were wound breakdown. In this contex, we included only the major complication (Grad III according to Calvin- Dindo classification) as mentioned above.

Our study was designed to include other complications.

  • Which did variables determine the choice of reconstruction?

We mentioned already, that (166-167) defect location, shape, and volume were all taken into consideration when deciding how to proceed with reconstructive surgery.

  • Authors stated that “In two individuals, the pelvic floor was reconstructed following a posterior exenteration by employing the corpus uteri as a muscle flap (Figure 1).” The employment of corpus uteri as a muscle flap is unclear in terms of safety, efficacy, and surgical technique. Can authors share more details citing the relevant references?

Using this flap is an innovative technique and a simple method to reconstruct perineal defect following extralevator abdominoperineal resection in women. We added the appropriate citation about this flap.

  • Did authors routinely use intraoperative indocyanine green angiography for each case of vulva reconstruction?

We mentioned in methods:

Throughout the past four years, intraoperative indocyanine green angiography (SPY-Portable Handheld Imager, SPY-PHI, Stryker, Kalamazoo, Michigan, US) was utilized to visualize blood flow in arteries and concomitant tissue perfusion (2018-2021). (Figure 2 depicts a fluorescence imaging system perfusion assessment one week following surgery after indocyanine green injection.)

Now, we use it in every case of reconstructive surgery.

  • The flap showed in the figure n.2 seems to be completely necrotic according both to the fluorescence and clinical picture taken one week after surgery. This issue should be indicated in the caption

We added this sentence to capture:
This figure showed the very poor flap perfusion indicated with ICG- imaging study intraopera-tively and the necrotic flap one week later.

  • The authors should indicate postoperative management of these patient, if any.

In results we mentioned:
There was only one incidence of total flap loss in the RS group (figure 2 depicts the in-traoperative assessment of flap perfusion with indocyanine green (a, b, c) and the post-operative ischemic flap(d)), and all other occurrences of wound healing disorder in the recipient site were wound breakdown. In these cases, the second plastic surgery with new reconstruction were indicated.

In the “Results”

  • How many flaps were transplanted in the study? Which flaps were employed according to the type of excisional surgery (anterior vulvectomy, radical vulvectomy, posterior vulvectomy, hemi-vulvectomy, exenteration)?

We perfomed 158 flaps, 104 as one flap per patients, in 12 patients 2 flaps and in 10 Patients 3 flaps.

The types of flaps and excisional surgeries were mentioned in details in table 2.

  • Information about preoperative radiotherapy should be given, because this factor has a fundamental role in delayed wound healing and postoperative complications rate.

There is no preoperative neoadjuvant radiotherapy, as it is not a standard in our hospital.

  • It is unclear the number of cases of total urethral resection. How did authors restore the case of partial urethral resection?

We mentioned that (158-160) A permanent supra-pubic catheter was placed in two cases with total urethral resection, and in the third case, complete bladder resection with ileum conduit was the treatment of choice. We do the partial resection if we know only max. 1 cm of urethra has to be resected. In these cases, there is no need for urethra reconstruction. If we need to resect more than 1 cm of urethra, then we resected the urethra for functional and oncological purposes completely and induce a suprapubic catheter if the bladder spared.

  • In which case did authors use myo-cutaneous flap? Which was the entity of the defect?

The myocutaneous flaps used for great defects or when the defect spread on more than functional and anatomical region (Vulva and groin or vulva and thigh).

  • In figure 2, pictures revealed an intraoperative ischemic aspect of the flap using indocyanine green angiography. Why did authors decide to not harvest a second flap?

This was at the beginning of using ICG for this purpose. Clinically the flap looked good perfused so we decided to take the risk and to keep it because it was a big flap with a large defect in the recipient and donor site. We published this figure because we want to show even the negative results and to encourage colleagues to take even such a difficult decision intraoperatively, if it is necessary.

  • Authors should indicate how many patients with postoperative wound healing disorders needed a second-stage surgery.

We mentioned already that (176-178) The wound healing disorder (Grad III according to Calvin-Dindo classification [14]) was registered in 16 patients (12.7%) of RS group vs. 7 patients of PC group (13.7%), p=1.000.

Grad III are defined as complications requiring surgical intervention.

In the Discussion:

  • Authors should indicate the strength and novelty of this retrospective study compared with others.

We added:

The strength of our study, that it includes, to the best of our knowledge, the largest number of patients underwent reconstructive surgery after excisions of vulvar cancer. All surgeries were performed from the same team omitting the selection bias induced by different surgical teams.

  • The importance of radiotherapy on complication rate should be better discussed, citing the relevant reference (PMID: 34830901)

Using the radiotherapy in a neoadjuvant setting is not the standard of care in our institution, as we aimed always to resect the tumor primarily and to induce the radiotherapy for risk patients (mainly in groin region but in vulvar region). Our study does ot include any patient after neoadjuvant radiotherapy.

  • The possibility of vulvoperineal reconstruction with perforator flaps should be mentioned and better discussed, citing the relevant references (PMID: 34866009; PMID: 27273808)

We added:

In this study, we do not discuss the wide options of using the perforator flaps for reconstructing the vulvoperineal defects, as our study do not include such options. Quite apart from the fact that these flaps are very effective having the advantages of low complication rate, minimal donor site morbidity, quick dissection, proximity of donor and recipient sites and the possibility to harvest large skin islands of variable thickness [20, 21].

Round 2

Reviewer 1 Report

The authors have done a decent job in this paper with respect to this being a retrospective analysis. They have made it clear the need for this article and what are its strong and weak points based on the patient data and also give a clear picture about reconstructive surgery versus primary closure in vulvar cancer. 

Reviewer 2 Report

The authors improved the quality of the manuscript taking into account the reviewers' comments. I feel that the paper is worthy of publication in the present form.